# Oxidative Phosphorylation Dysfunction Modifies the Cell Secretome

**DOI:** 10.3390/ijms21093374

**Published:** 2020-05-10

**Authors:** Nuria Garrido-Pérez, Ana Vela-Sebastián, Ester López-Gallardo, Sonia Emperador, Eldris Iglesias, Patricia Meade, Cecilia Jiménez-Mallebrera, Julio Montoya, M. Pilar Bayona-Bafaluy, Eduardo Ruiz-Pesini

**Affiliations:** 1Departamento de Bioquímica, Biología Molecular y Celular, Universidad de Zaragoza, C/Miguel Servet, 177, 50013 Zaragoza, Spain; anavela55@hotmail.com (A.V.-S.); esterlop@unizar.es (E.L.-G.); seortiz@unizar.es (S.E.); eiglesia@unizar.es (E.I.); pmeade@unizar.es (P.M.); jmontoya@unizar.es (J.M.); pbayona@unizar.es (M.P.B.-B.); 2Instituto de Investigación Sanitaria (IIS) de Aragón, Av. San Juan Bosco, 13, 50009 Zaragoza, Spain; 3Centro de Investigaciones Biomédicas en Red de Enfermedades Raras (CIBERER), Av. Monforte de Lemos, 3-5, 28029 Madrid, Spain; cjimenezm@fsjd.org; 4Instituto de Biocomputación y Física de Sistemas Complejos (BIFI), Universidad de Zaragoza, C/Mariano Esquillor (Edificio I + D), 50018 Zaragoza, Spain; 5Unidad de Patología Neuromuscular, Departamento de Neuropediatría, Institut de Recerca Sant Joan de Déu, Hospital Sant Joan de Déu, Passeig de Sant Joan de Déu, 2, 08950 Esplugues del Llobregat, Barcelona, Spain; 6Fundación ARAID, Av. de Ranillas, 1-D, 50018 Zaragoza, Spain

**Keywords:** oxidative phosphorylation system, mitochondrial DNA, mitochondrial diseases, secretome, biomarkers, fibroblast growth factor 21, growth differentiation factor 15, vascular endothelial growth factor, interleukine-6

## Abstract

Mitochondrial oxidative phosphorylation disorders are extremely heterogeneous conditions. Their clinical and genetic variability makes the identification of reliable and specific biomarkers very challenging. Until now, only a few studies have focused on the effect of a defective oxidative phosphorylation functioning on the cell’s secretome, although it could be a promising approach for the identification and pre-selection of potential circulating biomarkers for mitochondrial diseases. Here, we review the insights obtained from secretome studies with regard to oxidative phosphorylation dysfunction, and the biomarkers that appear, so far, to be promising to identify mitochondrial diseases. We propose two new biomarkers to be taken into account in future diagnostic trials.

## 1. Introduction

The secretome represents proteins that are secreted by a cell. Secretory proteins play important roles in communication between cells and, as a consequence, may coordinate and regulate biological activities [1,2,3]. Secretory factor-mediated signal transduction determines physiological processes such as proliferation, growth, differentiation, migration, and metabolic regulation [4]. Furthermore, because these secretory proteins are released into blood plasma, they are widely accepted to play important roles in biological responses and homeostasis of the whole body and are closely related to disease development including metabolic and neural diseases [5]. Secretome studies are at the focus of understanding physiological or pathological conditions. The identification of circulating proteins capable of being routinely used for diagnosis, prognosis, risk stratification, and therapeutic monitoring is currently of great interest [6]. Secretome analyses have been previously carried out for many pathophysiological conditions such as hypoxia, diabetes, and anti-cancer drug treatment [7,8,9], but very few of these are related to mitochondrial oxidative phosphorylation (OXPHOS) system diseases.

## 2. Oxidative Phosphorylation System 

Mitochondria play important roles in cellular energy production, metabolism, and cellular signaling [10,11,12]. Mitochondrial bioenergetics touch nearly every aspect of the cell. In addition to providing most of the energy, the mitochondria generate reactive oxygen species (ROS). Given the role that mitochondria play in a great number of cellular processes, it is not unsubstantiated to think that mitochondrial function and dysfunction could result in modifications of the cell secretome.

Over 1500 proteins are required for normal mitochondrial function [13,14]. Of these, only approximately 90 proteins are directly involved in the electron transport chain (ETC) and the production of adenosine triphosphate (ATP), i.e., the OXPHOS system. OXPHOS requires the transport of electrons to molecular oxygen by the mitochondrial respiratory chain which involves four multi-subunit complexes known as complexes I, II, III, and IV (CI, CII, CIII and CIV) and two mobile electron carriers, coenzyme Q (CoQ) and cytochrome *c* (Figure 1). The respiratory chain generates a transmembrane proton gradient that is channeled by complex V (also known as ATP synthase, CV) to synthesize ATP [15]. 

Almost all of the cell’s redox reactions ultimately feed electrons into the respiratory chain. CI and CII mediate the electron transfer from NADH and FADH_2_, respectively, to CoQ. CIII receives electrons from reduced CoQ and funnels electrons to cytochrome *c*. CIV ends the respiratory chain by accepting electrons from cytochrome *c* and using them to fully reduce oxygen to water [15]. The mitochondrial respiratory chain is the main source of ROS, particularly by CI and CIII (Figure 1). Excessive ROS might damage lipid membranes, proteins, and nucleic acids and have a role in the pathogenesis of mitochondrial diseases [10]. 

Mitochondria contain their own DNA, the mitochondrial DNA (mtDNA). Human mtDNA encodes 13 structural protein subunits of the OXPHOS system, and 2 ribosomal RNAs (rRNAs) and 22 transfer RNAs (tRNAs) that are required for intra-mitochondrial protein synthesis [16,17]. 

Apart from the 13 proteins encoded by the mtDNA, the rest of the mitochondrial proteins are encoded by the nuclear genome. These are translated in the cytoplasm and imported into the mitochondria. The OXPHOS system assembly requires the presence of tens of different factors. Altogether, well over 100 genes govern the process of oxidative phosphorylation, and mutations in any of these genes can cause an OXPHOS defect [17,18]. 

## 3. OXPHOS Dysfunction and Disease

Defective OXPHOS function results in disease. Mitochondrial diseases are the most common form of inherited metabolic disorders [10]. The pathophysiology of mitochondrial diseases is complex and involves genetic mutations in mtDNA or nDNA. In patients with mtDNA mutations, inheritance and clinical presentation are further complicated by the presence of multiple mtDNA genomes in an individual cell, which can often lead to a mixture of mutated and wild-type genomes, called heteroplasmy. The level of heteroplasmy is crucial in determining the extent of cellular dysfunction. Conventionally, mitochondrial diseases are a consequence of a primary defect in oxidative phosphorylation, the process by which cells produce ATP [10]. To further complicate the issue, acquired conditions, e.g., exposure to chemicals, can also lead to OXPHOS dysfunction. 

Mitochondrial diseases are clinically heterogeneous, can occur at any age, and can manifest with a wide range of clinical symptoms. Mitochondrial diseases can also involve any organ or tissue and characteristically involve multiple systems, typically affecting organs that are highly dependent on aerobic metabolism, and are often relentlessly progressive with high morbidity and mortality [19]. The heterogeneity in the clinical manifestation of mitochondrial diseases means that both diagnosis and management of these disorders are extremely difficult. Diagnosis often relies on genetic testing, in addition to histochemical and biochemical analysis of tissue biopsies. Establishing the molecular mechanisms that are responsible for the exceptional variability and tissue specificity of disease manifestations remains challenging [10]. 

In addition to the modifications in the local tissue environment, it is feasible that metabolic alterations in the tissues affected by mitochondrial dysfunction also reshape global metabolic signals at the whole-organism level. In this case, secreted molecules could influence how disease manifest in other tissues and potentially serve as biomarkers obtained from the peripheral blood [18]. 

## 4. OXPHOS Dysfunction Modifies the Protein Secretion by the Cells

Transmitochondrial cell lines called cytoplasmic hybrids, or cybrids, can be used to confidently link a phenotype to mtDNA mutations. These cells share nDNA and differ in their mtDNA. Prigione and Cortopassi [20] used cybrids of osteosarcoma 143B cells bearing mtDNA deletions and found that these deletions decreased cellular ATP production and the secretion of fibronectin (FN) and osteoprotegerin (OPG). As a positive control they included a rho zero (rho^0^) cell line, experimentally depleted of mtDNA, which showed the same results. As negative control cells, they used cells that had been cybridized with nonpathogenic mtDNAs. In the same study, CI inhibitor rotenone (Figure 1) recapitulated the decrease in ATP production as well as the inhibition of synthesis and secretion of FN and OPG, suggesting that these are all consequences of decreased energy status [20]. 

Other genetic manipulations that impaired the OXPHOS system, also modify the protein secretion by the cells. A knock-in mouse for mutated thymidine kinase 2 (TK2), a deoxyribonucleoside kinase required for mtDNA synthesis, showed mtDNA depletion in white adipose tissue accompanied with reduced fat accumulation. These mice also showed a severe reduction in mRNA and circulating protein levels of leptin, an adipose-derived hormone involved in the regulation of food intake and energy expenditure [21]. Mouse 3T3-L1 cells knocked down for the mitochondrial transcription factor A (TFAM), a key activator of mitochondrial transcription as well as a participant in mitochondrial genome replication, showed a decrease in mtDNA copy number, levels of ETC subunits, CI and CIV activities, and oxygen consumption. These cells also showed a diminished expression at both mRNA and secreted protein levels of another hormone, adiponectin, that regulates glucose levels and fatty acid catabolism [22]. Adiponectin mRNA expression was also decreased in TFAM knocked down human mesenchymal stem cells (hMSCs) [23]. Mouse 3T3-L1 or adipose-tissue-derived stem cells (mASCs) that lacked the growth arrest and DNA damage-inducible proteins-interacting protein 1 (GADD45GIP1), a mitochondrial translation/assembly factor for mtDNA-encoded polypeptides, expressed lower levels of mtDNA-encoded subunits and resulted in profound impairment of OXPHOS function. The cells displayed disrupted adipocyte differentiation, accompanied by a reduced adiponectin expression [24]. Fibroblast growth factor 21 (FGF21) and angiopoietin-like 6 (ANGPTL6), were highly induced in another model of homozygous GADD45GIP1-deficient mouse embryonic fibroblasts. This group later generated adipocyte-specific GADD45GIP1-deficient mice (AdKO). Adipose tissue from AdKO mice displayed significantly reduced levels of OXPHOS subunits compared to that from the control group. Protein expression of ANGPTL6 and FGF21 was also higher in the white adipose tissue and brown adipose tissue of AdKO mice as were their levels in serum [25].

Chemical manipulation of the OXPHOS system can also affect protein secretion (Figure 1). CI inhibitor rotenone decreased adiponectin secretion in rat adipocytes and adiponectin mRNA expression in hMSCs [23,26]. Capsaicin, another CI inhibitor, decreased leptin and increased adiponectin expression in 3T3-L1 adipocytes [27]. The CIII inhibitor antimycin A reduced adiponectin mRNA levels in mouse 3T3-L1 cells [28]. Finally, the CV inhibitor oligomycin diminished adiponectin mRNA levels and secreted adiponectin in mouse 3T3-L1 cells [22,28].

Some therapeutic drugs that impair OXPHOS function are widely used in humans. This is the case for nucleoside reverse-transcriptase inhibitors (NRTIs) used against the human immunodeficiency virus (HIV) which can inhibit the mitochondrial polymerase gamma (POLG), responsible for mtDNA replication. NRTIs decreased adiponectin mRNA expression and protein secretion in mouse 3T3-L1 and 3T3-F442A cells and in primary human subcutaneous preadipocytes [29,30,31]. Systemic adiponectin levels were also reduced in patients under antiretroviral therapy [32]. Our group studied the effect of NRTIs on the OXPHOS system during differentiation of human adipose-derived stem cells (hASCs). The experiments showed that 2′,3′-dideoxycytidine (ddC), a specific inhibitor of POLG mtDNA synthesis, reduced adipocyte differentiation and leptin secretion by hASCs-derived adipocytes and this was accompanied with significantly reduced mtDNA levels [33]. Linezolid (LIN) is a ribosomal antibiotic with effects on mitochondrial translation and OXPHOS function. In our hands, at concentrations below the steady-state peak serum concentrations [34], LIN decreased CIV activity and inhibited mitochondrial protein synthesis in hASCs. LIN also impaired differentiation of hASCs as indicated by decreased intracellular triglycerides (TGs) and secreted leptin [33]. In agreement, chloramphenicol (CAM), another mitochondrial translation inhibitor, tends to decrease the levels of secreted leptin as well [33]. These results indicate that the OXPHOS dysfunction modifies the adipokine secretion. 

## 5. OXPHOS Dysfunction and Secretomic Studies

In view of the above-mentioned results, we analyzed the effect of LIN on the whole adipocyte secretome by a mass-spectrometry-based secretome profiling. 48 h serum-free cell culture medium was collected and processed according to [35,36]. Samples were analyzed by nano-liquid chromatography coupled with an ion-trap mass spectrometer [37]. Following this procedure, the culture and adipocyte differentiation of hASCs in the presence or absence of LIN yielded changes in thirteen secreted proteins. Surprisingly, we did not find adiponectin and leptin in the secretome analysis by mass spectrometry despite the fact that, using enzyme-linked immunosorbent assay (ELISA), we previously observed a 4- and 6-fold increase, respectively, in their culture medium levels [33]. However, leptin is mainly detected with antibody-based methods and its absence is not a rare observation for this kind of analysis [38]. We focused in four proteins not related with cytoskeleton or extracellular matrix: fatty acid-binding protein 4 (FABP4), apolipoprotein E (APOE), plasminogen activator inhibitor 1 (PAI-1), and complement factor D (adipsin). FABP4 and APOE levels increased with normal adipocyte differentiation, but LIN partially counteracted this increase. APOE was found specifically in the cell culture medium of adipocytes and was not present in hASCs. Adipsin amount also increased during differentiation and this was potentiated by LIN. PAI-1 quantity decreased over the adipocyte differentiation, but LIN partially inhibited this drop. Fibronectin (FN) secretion was decreased after adipocyte differentiation. This reduction was somewhat mitigated when differentiation was performed in the presence of LIN [39]. FN secretion was earlier found to be decreased in cybrid cells bearing mtDNA deletions and showing a reduced ATP production [20]. The cell lines used in the two experiments (143B cells vs. adipocytes) and the proliferating or differentiating stages could explain the discrepancy between these results. 

Defects in OXPHOS function may result in pathological production of ROS. Conversely, ROS overproduction can lead to OXPHOS dysfunction [40]. Meyer, at al. [41] studied the effect of ROS on human stem-cell-derived retinal pigmented epithelium (RPE) secretome. RPE samples were stressed with paraquat (PQ), an inducer of mitochondrial ROS production, and the levels of mitochondrial CI and CII subunits were decreased. The secreted proteome was quantified by mass spectrometry. Twenty-four-hour serum-free cell culture medium was processed and analyzed by nanoflow liquid-chromatography-tandem mass spectrometry analysis (nano-LC-MS/MS). They found that ROS decreased FN, complement cascade factors, and APOE secretion. Growth differentiation factor 15 (GDF15) was highly upregulated [41]. Noteworthy, FN, APOE, and several factors of the complement secretion were modified in secretome analysis under OXPHOS dysfunction conditions [39,41]. 

Besides the last two studies mentioned above, no other secretomic analyses related to OXPHOS dysfunction have been published so far, although studies of specific proteins secreted by the cells under several conditions affecting OXPHOS function are mentioned later.

## 6. Biomarkers for OXPHOS Dysfunction

Biomarkers are indicators of biologic or pathogenic processes used for disease diagnosis, for monitoring the disease progression, and for patient response to therapeutic interventions. Since diagnosis of mitochondrial disorders is challenging, the identification of easily-accessible biomarkers is of utmost importance. Because OXPHOS dysfunction has the capacity to modify the secretome, some effort has been carried out to discover secreted mitochondrial disease-specific biomarkers or regulators of secretory signals [5]. However, given the complexity of the OXPHOS diseases, it is likely that the future diagnostic tool will not rely on a single biomarker alone but a combination of many, a “biosignature”, is required [42].

In the past years, serum FGF21 and serum GDF15 have emerged as two promising diagnostic biomarkers for mitochondrial diseases [18,43,44,45,46,47]. They have been subsequently validated in patient cohorts [48]. 

## 7. Fibroblast Growth Factor 21

FGF21 is a member of the fibroblast growth factor superfamily. Circulating FGF21 in humans derives mainly from the liver, but is also known to be secreted by adipocytes, myocytes, and the pancreas [49]. FGF21 is a hormone-like cytokine known to play key roles in glucose and lipid metabolism and it is currently being pursued as a therapeutic drug for obesity and T2-diabetes [50]. FGF21 has been associated with anti-inflammatory properties in various tissues [51].

FGF21 has been found to be related to OXPHOS dysfunction, as it was induced in a late-onset mitochondrial myopathy mouse model, a transgenic mouse expressing a mutation in the twinkle mtDNA helicase (TWNK) [52]. Transcriptomic analysis in this model indicated that respiratory chain-deficient skeletal muscle initiated a specific pseudo-starvation response, which was associated with considerable changes in the serum amino acid, lipid, and cytokine levels [53]. Interestingly, the serum response followed closely the progression of the pathological findings in the skeletal muscle and responded to treatment; the serum amino acid profile and cytokine levels were normalized after ketogenic diet, along with the reduction in ultrastructural mitochondrial abnormalities [53,54]. Subsequently it was confirmed that blood FGF21 levels were highly increased in patients with primary muscle-manifesting respiratory chain deficiencies, mostly those caused by pathogenic mutations in mitochondrial DNA, correlated with disease severity and respiratory chain-deficient muscle fibers [18,55]. Therefore, serum FGF21 was reported as a potential biomarker for mitochondrial diseases [18].

The increased FGF21 secretion observed in the muscles of patients suffering from mitochondrial myopathy was recapitulated in vitro with mitochondrial respiratory chain inhibitors. FGF21 secretion was increased in differentiated mouse C2C12 myotubes treated with an inhibitor of complex II (TTFA, thenoyltrifluoroacetone), a respiratory chain uncoupler (FCCP, carbonyl cyanide 4-[trifluoromethoxy] phenylhydrazone), or the complex IV inhibitor sodium azide in a dose-dependent manner [55] (Figure 1). 

Recently, it was found that the highest induction of FGF21 response occurs in mitochondrial disorders that primarily or secondarily affect mitochondrial translation, such as direct mutations of translation machinery or mtDNA deletions leading to imbalance of mtDNA-encoded tRNAs and rRNAs but not mutations in structural respiratory chain complexes or their assembly factors [56]. mtDNA maintenance disorders, such as those caused by nuclear gene mutations in POLG, TWNK, or thymidine phosphorylase (TYMP), also caused mtDNA deletions or point mutations and induced FGF21 [57,58,59]. Consistent with human data, mouse models accumulating multiple mtDNA deletions in skeletal muscle [52,60], or those with a single large heteroplasmic mtDNA deletion [61], induced FGF21, clearly linking mtDNA deletions to the cytokine response [56].

## 8. Growth Differentiation Factor 15

GDF15 is a cytokine of the transforming growth factor β (TGF-β) superfamily, which is expressed mainly in the placenta, kidney, liver, lung, pancreas, and prostate [62,63,64]. Expression of GDF15 in nearly all tissues suggests its general importance in essential cellular functions [65]. It has an essential role in regulating the cellular response to stress signals and inflammation, being involved in suppression of inflammation in early pregnancy, cancer, and cardiovascular diseases. Moreover, GDF15 is expressed in the choroid plexus acting as a potent neurotrophic factor for motor and sensory neurons [64]. 

We identified GDF15 as a potential diagnostic biomarker for mitochondrial diseases by a gene expression study in TK2-deficient human skeletal muscles [45]. GDF15 mRNA levels were dramatically increased in muscle from patients with TK2 mutations and the protein was constitutively secreted by skeletal muscle cells. 

In order to validate the feasibility of GDF15 as a serum biomarker, its concentration was measured in the serum of 17 patients with mitochondrial diseases as well as in that of 13 patients with other pediatric diseases as a control [46]. GDF15 levels were significantly increased in the serum of mitochondrial disease patients and could clearly distinguish mitochondrial disease patients from control patients. The value of GDF15 for evaluating the therapeutic efficacy of pyruvate was remarkable [46,66]. 

Later [47] it was found that GDF15 was significantly elevated in mitochondrial disease patients, and appeared to increase with the clinical severity of the disease. In this study, mean GDF15 levels were ranked in the following order: Leigh syndrome (LS); mitochondrial encephalopathy, lactic acidosis, and stroke-like episodes (MELAS); Kearns–Sayre syndrome (KSS); overlapping MELAS/LS; and mitochondrial encephalopathy and lactic acidosis (MELA).

## 9. Fibroblast Growth Factor 21 vs Growth Differentiation Factor 15

Many biomarker studies in the last few years have been focused on both FGF21 vs GDF15 simultaneously, comparing their sensitivity and specificity as mitochondrial biomarkers. They were tested in a large patient cohort with different mitochondrial defects and it was found that GDF15 showed a higher sensitivity and specificity than FGF21 [47]. We have subsequently compared circulating GDF15 and FGF21 levels in a cohort consisting exclusively of children with a diagnosis of mitochondrial disease which included patients with mutations in both mtDNA and nDNA [67]. The results indicate that GDF15 is a sensitive and specific biomarker to guide the diagnosis of this group of complex genetic diseases. Furthermore, it was shown that the combined use of GDF15 and FGF21 was more efficient in identifying patients than either factor alone. This strategy would be useful for example to select patients for comprehensive genetic analysis, which is still expensive and not available in all centers. 

Considering the above data in patients, we determined whether mitochondrial dysfunction affects GDF15 gene expression in muscle cells [67]. To this end, C2C12 myotubes were treated with antimycin A or oligomycin (CIII and CV inhibitors, respectively, Figure 1) which caused a dramatic induction of GDF15 gene expression, as well as of FGF21 gene expression. In parallel experiments, the effects of antimycin A and oligomycin were determined in human LHCN-M2 myotubes. Similar induction was observed in human LHCN-M2 myotubes under the same conditions. Previous studies have shown that ROS production was involved in the induction of FGF21 expression by experimental mitochondrial dysfunction [68]. Effectively, treatment of C2C12 myotubes with the ROS scavenger, Trolox, blunted antimycin- and oligomycin-induced FGF21 expression. In contrast, the induction of GDF15 by the mitochondrial function inhibitors was insensitive to the presence of the ROS scavenger [67]. As mentioned above, GDF15 was highly upregulated after chronic ROS in a human stem cell-derived RPE model [41], pointing to different regulatory pathways or experimental conditions acting in the two models [41,67]. However, as with FGF21 [69], it was shown that serum GDF15 does not correlate with disease severity in a large cohort of adult m.3243A>G mtDNA mutation carriers [70]. On the other hand, GDF15 seems to be more indicative of mitochondrial diseases regardless of clinical phenotype, whereas FGF21 sensitivity for mitochondrial diseases is higher when muscle manifestations are present [45]. It seems that GDF15 levels in serum are higher in mitochondrial disease patients with multisystem involvement, like MELAS or Pearson/KSS patients [47]. Likewise serum FGF21, serum GDF15 seems to be a more specific marker for mitochondrial diseases due to mitochondrial translation and mtDNA maintenance defects, as opposed to those resulting from impaired respiratory chain complex or assembly factors [71]. Similarly to FGF21, the reliability and efficacy of GDF15 as a biomarker of mitochondrial diseases remains to be tested in others patient cohorts [42].

Presently, both FGF21 and GDF15 are more sensitive and specific than currently-used clinical diagnostic markers of mitochondrial disorders such as lactate, pyruvate, creatine kinase, alanine, or organic acids [47,67,71]. However, they are yet to be incorporated into formal diagnostic pathways. It is possible that future studies will identify additional biomarkers that, together with or separately from these two, may help in the diagnosis of mitochondrial diseases. 

## 10. Vascular Endothelial Growth Factor

Another secreted protein which is modified by OXPHOS dysfunction is VEGF. VEGF represents a family of signaling proteins involved in both vascular development and angiogenesis [72]. Increased VEGF mRNA expression, and stimulation of the angiogenic pathway, were found in paraganglioma and pheochromocytoma tumors carrying mutations in nDNA-encoded CII subunits SDHB and SDHD that result in decreased CII activity [73,74]. Similarly, increased VEGF mRNA and protein expression were found in human embryonic kidney HEK293 cells carrying mutations in the nDNA-encoded CIII subunit UQCRB with decreased OXPHOS function [75]. Human SK-Hep1 hepatoma rho^0^ cells express more VEGF mRNA and protein than parental cells with mtDNA, rho^+^ cells. Conditioned medium from these rho^0^ cells increased the formation of tube-like structures from human umbilical vein endothelial cells and new blood vessels in chorioallantoic membrane assays [76]. An inducible mtDNA-depletor mouse expressing a dominant-negative mutation in the polymerase domain of POLG, induced mtDNA depletion in various tissues. These mice showed reduced mtDNA content, reduced mitochondrial gene expression and OXPHOS enzyme activities, and increased expression of VEGF mRNA, especially in skin [77]. Cybrids obtained from mouse Lewis lung carcinoma cells, harboring a CI mutation and having an OXPHOS defect, showed increased VEGF mRNA and protein levels and higher ability to induce neoangiogenesis than those with no mtDNA mutation [78,79].

Given these previous observations, OXPHOS inhibitors were used to cause mitochondrial dysfunction and VEGF levels were analyzed (Figure 1). In accordance, in vitro pretreatment of human adipose-derived stroma cells with CI or CIII inhibitors (rotenone or antimycin, respectively) increased VEGF secretion [80]. Another CI inhibitor (1-methyl-4-phenyl-1,2,3,6-tetrahydropyridine, MPTP), frequently used to model Parkinson disease, also increased the number of VEGF-expressing neurons and blood vessels in the substantia nigra of parkinsonian-rendered monkeys [81]. Administration of CIV inhibitor, sodium cyanide, to human brain microvascular pericytes resulted in increased expression of VEGF, and hydrogen sulfide, another CIV inhibitor, also has proangiogenic effects [82,83]. The ATP synthase inhibitor oligomycin increased VEGF protein production in the human U-937 monocytic cell line [84]. Likewise, 4-hydroxy-2-nonenal (4-HNE), an oxidative stress inducer, increased mtDNA point mutations and reduced CIII and CIV activity and oxygen consumption in primary rheumatoid arthritis synovial fibroblasts (RASF). 4-HNE also increased RASF VEGF immunofluorescence staining and VEGF secretion. The number of tube-like structures produced by human umbilical vein endothelial cells was also increased by 4-HNE RASF-conditioned medium [85]. 

The above data indicates that OXPHOS dysfunction increases VEGF expression and secretion as well as pathological angiogenesis, which is a feature of many OXPHOS disorders. Therefore, VEGF might be tested as a potential biomarker for certain mitochondrial diseases.

## 11. Interleukin-6

IL6 is a well-known myokine [5] as well as a pro- and anti-inflammatory cytokine [86]. Changes in IL6 secretion have been observed after manipulations that compromise the OXPHOS function. MSCs obtained from atherosclerosis subjects have greater levels of mitochondrial ROS and oxidative stress than non-atherosclerosis MSCs. Atherosclerotic-MSCs display a higher reduction of mitochondrial membrane potential, mitochondrial respiration (OCR: key metric of OXPHOS), and sensitivity to electron transport chain inhibitors than nonatherosclerotic-MSCs. Atherosclerotic-MSCs have also decreased levels of CI subunit NDUFB8, CII subunit SDHB, CIII subunit core 2, and CIV subunit p.MT-CO2 [87]. Despite being cultured under normoxic conditions, atherosclerotic-MSCs have increased hypoxia-inducible factor 1α (HIF-1α) protein levels. Given the higher abundance of HIF-1α in atherosclerotic-MSCs, key cytokines in the MSC medium were evaluated. Atherosclerotic-MSCs secrete higher levels of IL6, C-X-C motif chemokine ligand 8 (CXCL8), and monocyte chemoattractant protein-1/C-C motif chemokine ligand 2 (CCL2) than nonatherosclerotic-MSCs in both resting and primed conditions. Moreover, CV inhibitor oligomycin-induced mitochondrial dysfunction (Figure 1) of nonatherosclerotic-MSCs leads to an increase in ROS levels and to a shift in the cytokine/chemokine secretome similar to atherosclerotic-MSCs. Of relevance, oligomycin treatment had no impact on the atherosclerotic-MSC secretome [87]. To note, when atherosclerotic-MSCs were treated with the ROS scavenger N-acetyl-l-cysteine and measured key cytokines and chemokines of the MSC secretome, the levels of IL6, CXCL8, and CCL2, were diminished, and the atherosclerotic-MSCs survival and immunopotency were enhanced [87]. Furthermore, the mitochondria-targeted ROS scavengers, Trolox, MitoCP, and MitoTempo similarly improved the atherosclerotic-MSCs immunomodulatory capacity. Therefore, it seems that the impaired mitochondrial function of atherosclerotic-MSCs underlies their altered secretome and reduced immunopotency.

In a Kupffer-cell model of trauma-hemorrhage, the ATP levels were decreased and this was accompanied by a decline in TFAM and mtDNA-encoded p.MT-CO1 from CIV [88]. Also, IL6 and TNF-α production capacities were increased following trauma-hemorrhage. Administration of 17ß-estradiol following trauma-hemorrhage increased ATP levels and normalized Tfam, *MT-CO1* mRNA and p.MT-CO1 levels as well as IL6 and TNF-α production capacity. These results suggest that 17ß-estradiol-upregulated ATP production and improved OXPHOS function in Kupffer cells leads to downregulation of cytokine release following trauma-hemorrhage.

In other experiments carried out, C57BL/6J (B6)-mice-derived Lewis lung carcinoma rho^0^ P29 cells were cybridized with mtDNA from senescence-accelerated mice P1 carrying the m.11181A>G mutation in the *MT-ND4* gene. These cybrids overproduce ROS, and, when co-cultured with dendritic cells from B6 mice, induced IL6 secretion was observed from B6 cells [89]. Moreover, mutations in *POLG* gene (A467T and W748S) in Alpers disease patients lead to a fatal brain and liver mitochondrial depletion syndrome with reduced activity of respiratory chain enzyme complexes [90]. Later it was observed that Alpers patients have elevated IL6, CXCL8, and IFN-c levels in cerebrospinal fluid [91]. The above studies indicate that IL6 secretion increases along with mitochondrial OXPHOS dysfunction so IL6 could be tested as a potential biomarker for mitochondrial diseases.

## 12. Biosignatures 

FGF21, GDF15, as well as VEGF and IL6, have been associated individually with a range of non-mitochondrial diseases, encompassing cancer, obesity, renal disease, diabetes, and liver disease, although many of them are characterized by an OXPHOS dysfunction [92,93]. Additionally, serum FGF21 is variably increased in non-mitochondrial myopathies [56,94].

Moreover, the commonly-observed co-regulation of secretome proteins, as demonstrated above, is not only observed in the context of mitochondrial dysfunction. Parallel changes for secreted FGF21, GD15, VEGF, and IL6 proteins have been demonstrated under several physiological or pathological conditions. In a study to evaluate FGF21 as a marker of mitochondrial dysfunction in the context of acute-on-chronic liver failure, FGF21 levels and IL6 were found increased in peripheral blood samples of the patients [95]. Moreover, FGF21 and IL6 serum levels were higher in patients with ulcerative colitis and irritable bowel syndrome [96] and significantly elevated in patients with severe compared with mild acute pancreatitis [97]. Levels of GDF15 and IL6 have been found increased in acute heart failure patients [98], the two proteins were also increased in anemic T2 diabetes (T2D) patients compared with non-anemic T2D patients [99]. Finally, positive correlation was demonstrated between GDF15 and IL6 in patients suffering from early stages of chronic kidney disease [100]. Secretome proteins may also be coordinately regulated in response to environmental factors such as temperature and hazardous substances. For many years, the IL6 has been known to be secreted in response to cold environment and in response to noradrenaline in mouse brown adipocytes in primary culture [101]. Similarly, GDF15 is intensely secreted by brown fat upon exposure to cold and brown adipocytes after noradrenergic stimulation in mice [102,103]. Recently, FGF21 and IL6 were both found induced in a mouse model of cold-induced thermogenesis [104]. IL6, FGF21, and GDF15 levels were also elevated in patients with myocardial infarction treated with therapeutic hypothermia [105]. Finally, it was shown that nickel can induce the production of IL6 [106,107,108] as well as several other secreted signaling proteins including GDF15 and VEGF in keratocytes [109]. 

As these biomarkers seem to behave in a correlative manner in several disease and environmental conditions, we propose that their combination as a “biosignature” for mitochondrial diseases warrants further investigation (Figure 2).

## 13. Conclusions

Heterogeneity is a well-recognized feature of virtually all parameters associated with OXPHOS diseases. Some metabolites firmly established in the clinical investigation of mitochondrial diseases, such as lactate, have a poor diagnostic sensitivity and specificity [48]. Recent developments in this field point at FGF21 and GDF15 as more sensitive and specific than currently-used clinical diagnostic markers, but this needs to be confirmed in other and large patient cohorts before being incorporated into routine diagnostic trials. Moreover, several secreted proteins together would increase their value further and be adequate to define a mitochondrial disease fingerprint. None of the markers discussed here is a specific biomarker of OXPHOS disease. However, given the ease of their determination in blood, the levels of these secreted proteins could be useful, together with other clinical clues, to determine if a suspected patient should proceed to a more invasive procedure, such as a muscle biopsy, to study histochemical and biochemical markers of OXPHOS disease [110]. In addition, the determination of the secreted protein levels could be useful in patients already diagnosed with an OXPHOS disease to predict their prognosis and response to treatment [48]. Although the secretion of several proteins is modified in response to OXPHOS dysfunction there are no reports on the secretome of patients with OXPHOS diseases. Therefore, secretome studies on patients with confirmed OXPHOS disease are warranted. These studies should include patients with non-OXPHOS diseases with major involvement of the muscle and/or nervous system, the most affected tissues in mitochondrial diseases. Moreover, different technical approaches must be considered in order to increase the number of proteins whose secretion is modified upon OXPHOS dysfunction.

## Figures and Tables

**Figure 1 ijms-21-03374-f001:**
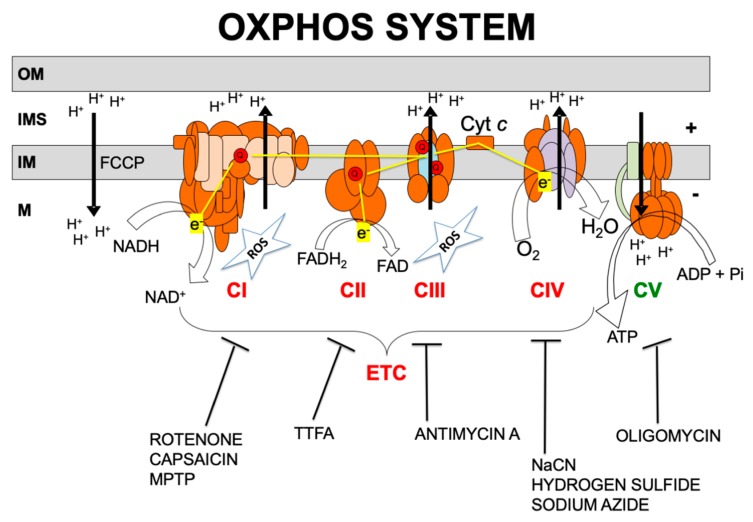
Oxidative phosphorylation (OXPHOS) system. OM, IMS, IM, and M code for mitochondrial outer membrane, intermembrane space, mitochondrial inner membrane, and mitochondrial matrix, respectively; ETC, electron transport chain; CI, CII, CIII, CIV, CV, and Cyt *c*, code for respiratory complexes I, II, III, IV, ATP synthase, and cytochrome *c*, respectively; Q, coenzyme Q_10_; NADH and NAD^+^, reduced and oxidized forms of nicotinamide adenine dinucleotide; FADH_2_ and FAD, reduced and oxidized forms of flavin adenine dinucleotide, H^+^, protons; e^−^, electrons; ATP, ADP, and Pi, adenosine triphosphate, adenosine diphosphate, and inorganic phosphate, respectively; H_2_O, water; O_2_, oxygen; ROS, reactive oxygen species. OXPHOS inhibitors: rotenone, capsaicin; MPTP: 1-methyl-4-phenyl-1,2,3,6-tetrahydropyridine; TTFA: thenoyltrifluoroacetone; antimycin A; NaCN: sodium cyanide; hydrogen sulfide; sodium azide; oligomycin; FCCP: carbonyl cyanide 4-[trifluoromethoxy] phenylhydrazone.

**Figure 2 ijms-21-03374-f002:**
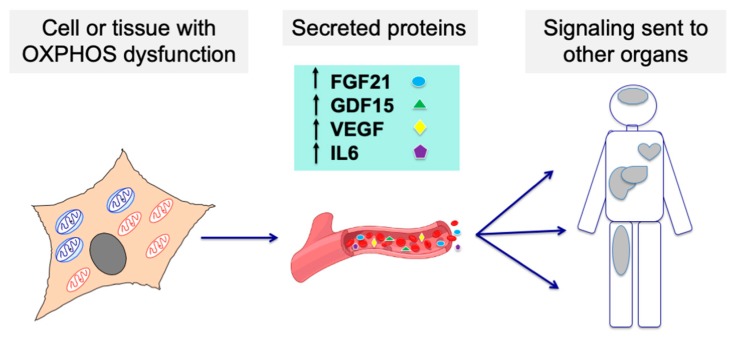
Cells or tissues affected by an OXPHOS dysfunction may modify their secretome, sending signaling molecules to other organs through the bloodstream and affecting their function. Such molecules could be detected and used as biomarkers for mitochondrial diseases.

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
