# Peer review of "Oxidative Phosphorylation Dysfunction Modifies the Cell Secretome"

_ijms, 2020, doi:10.3390/ijms21093374_

Round 1

Reviewer 1 Report

The manuscript from Garrido-Pérez et al. provides a detailed overview of the changes in cell secretome associated with defective oxidative phosphorylation, with the aim to identify reliable and specific circulating biomarkers for mitochondrial diseases. Furthermore, two new biomarkers  (VEGF and IL-6) are discussed.

This topic is of general interest for the implications on the clinical routine of mitochondrial diseases.

The manuscript is well organized and written, clearly describing the mitochondrial function and dysfunctions leading to mitochondrial diseases and the main cell secretome biomarkers identified so far.

I believe that the manuscript is suitable for publication with minor adjustments.

Specific comments

  1. The Authors should provide more details on the general experimental procedures used for determining the cell secretome (in cell lysates or in extracellular medium, time of measurement, type of mass spectrometry analysis, etc., lines 205-206). 
  2. The different biomarkers described in this study are potentially very interesting, however they are unfortunately present also in different non-mitochondrial diseases and environmental conditions, therefore the Conclusions should more strongly emphasize the need for further studies in a wider cohort of patients with different mitochondrial diseases before diagnostic use.

Line 152: a typo - rho cero  should be  rho zero cells

Author Response

Response to the reviewer 1.

The manuscript from Garrido-Pérez et al. provides a detailed overview of the changes in cell secretome associated with defective oxidative phosphorylation, with the aim to identify reliable and specific circulating biomarkers for mitochondrial diseases. Furthermore, two new biomarkers (VEGF and IL-6) are discussed.

This topic is of general interest for the implications on the clinical routine of mitochondrial diseases.

The manuscript is well organized and written, clearly describing the mitochondrial function and dysfunctions leading to mitochondrial diseases and the main cell secretome biomarkers identified so far.

I believe that the manuscript is suitable for publication with minor adjustments.

We would like to thank this reviewer for his/her kind comments and important suggestions.

Specific comments

  1. The Authors should provide more details on the general experimental procedures used for determining the cell secretome (in cell lysates or in extracellular medium, time of measurement, type of mass spectrometry analysis, etc., lines 205-206). 

We have now incorporated more details on the experimental procedures used in [1] and [2], the only ones reporting analyses of the secretome of cells with OXPHOS dysfunction. Lines 212-216 and 239-241.

[1] Proteome and secretome dynamics of human retinal pigment epithelium in response to reactive oxygen species. Meyer JG, García TY, Schilling B, Gibson BW, Lamba DA. Scientific Reports 2019; 9: 1-12.

[2] Pharmacologic concentrations of linezolid modify oxidative phosphorylation function and adipocyte secretome. Llobet L, Bayona-Bafaluy MP, Pacheu-Grau D, Torres-Pérez E, Arbones-Mainar JM, Navarro MA, Gómez-Díaz C, Montoya J, López-Gallardo E, Ruiz-Pesini E. Redox Biology 2017; 13: 244-254.

  1. The different biomarkers described in this study are potentially very interesting, however they are unfortunately present also in different non-mitochondrial diseases and environmental conditions, therefore the Conclusions should more strongly emphasize the need for further studies in a wider cohort of patients with different mitochondrial diseases before diagnostic use.

We agree with the reviewer. We have now extended the Conclusions section to emphasize the need for further studies in wider cohorts of patients with different mitochondrial diseases before diagnostic use. Lines 483-502.

     - Line 152: a typo - rho cero should be rho zero cells

We have now corrected this typo. Line 159.

Submission Date 18 March 2020 Date of this review 25 Mar 2020 17:40:42

Reviewer 2 Report

The authors contend that mt dysfunction, reflected in large part by congenital or acquired alterations in the OXPHOS apparatus, modify the cellular secretome in ways that may create useful diagnostic circulating biomarkers of mt disease. Ultimately, they decide 4 nuclear-encoded non-mt proteins, FDF21, GDF15, VEGF and IL6, may serve such a role, particularly if considered together. While it is reasonable to assume that, at least conceptually, certain secreted proteins or other factors may have diagnostic potential in specific mt disorders, such a case has not been made here, in my opinion. 

Besides the compounds noted above, the authors discuss such molecules as leptin, adiponectin, APOE and others, altered secretion of which have never been ascribed to a fundamental, primary defect in mt function, and certainly have not been studied prospectively as diagnostic markers in clinical trial involving primary mt diseases. However, much the same can be said for the 4 putative biomarkers mentioned above. FGF21 has been most studied and does appear to be related in some way to OXPHOS disorders. However, neither it nor the other listed compounds have been studied rigorously in randomized controlled trials (RCTs) of primary mt diseases sufficiently to allow the FDA or similar regulatory agencies to accept any as an a priori, valid biomarker that could be used in a RCT as a major outcome variable of a therapeutic intervention. The main objection is illustrated by the authors' own admission, summarized in lines 432-452, of the non-specificity of these molecules. As they state, many diseases, few (if any) of which are generally thought of as primarily acquired (much less congenital) mt diseases (eg., IBS, myocardial infarction), can elaborate 1 or more of these factors (FGF21, VEGF, etc.). Thus, what is their value as mt disease biomarkers when they can be shown to correlate with so many other conditions? 

Specific comments:

  1. The paper's Inroduction is ~50% too long; it currently takes4 pages and over 140 lines to reach the beginning of the title's real theme: "OXPHOS dysfunction modifies the protein secretion by the cells".
  2. Fig. 1 includes various OXPHOS inhibitors, several of which are discuss later in the paper. It would be useful to refer back to this figure to remind the reader where these poisons act. 
  3. Lines 123-124: this sentence reflects a tautology and can be eliminated. 
  4. line 125: a specific reference is required to justify this claim.
  5. lines 226-238: what is the relevance of this passage to the article's main theme? It can be severely shortened or eliminated.
  6. line 278: some of these compounds are not identified in Fig. 1.
  7. lines 316-17: either eliminate the word "remarkable" or defend its use with actual data.
  8. lines 353-354: this statement should  either be defended with data and references or eliminated. To what are the authors referring as "currently used clinical markers and how has their utility been compared to that of FGF21 and/or GDF15?
  9. lines 357-391: it seems extremely unlikely, and misleading, to promote VEGF as a future biomarker of OXPHOS dysfunction. A major HIF1a -induced transcript, VEGF is increased in so many physiological and patholigical conditions as to make it a  sensitive and specific marker, alone or with other molecules, of mt diseases extremely unlikely. A similar concern applies to IL-6. These concerns beg the question how the authors would design a study to determine, with confidence, the validity of using the 4 biomarkers listed in the figure for the diagnosis of diseases of OXPHOS disorders.
  10. Fig 2:VEGF is misspelled.

Author Response

Response to the reviewer 2.

The authors contend that mt dysfunction, reflected in large part by congenital or acquired alterations in the OXPHOS apparatus, modify the cellular secretome in ways that may create useful diagnostic circulating biomarkers of mt disease. Ultimately, they decide 4 nuclear-encoded non-mt proteins, FDF21, GDF15, VEGF and IL6, may serve such a role, particularly if considered together. While it is reasonable to assume that, at least conceptually, certain secreted proteins or other factors may have diagnostic potential in specific mt disorders, such a case has not been made here, in my opinion.

            First, we would like to thank this reviewer for his/her in-depth review and accurate comments.

            We agree with the reviewer and, possibly, we did not explain ourselves correctly. Our hypothesis, and the reviewer consider that it is reasonable to assume, is that “certain secreted proteins … may have diagnostic potential in specific mt disorders …”. What we wanted to hold up on our review is that secretome studies in patients suffering OXPHOS disorders may provide clues for the management of these patients (see below).

Besides the compounds noted above, the authors discuss such molecules as leptin, adiponectin, APOE and others, altered secretion of which have never been ascribed to a fundamental, primary defect in mt function, and certainly have not been studied prospectively as diagnostic markers in clinical trial involving primary mt diseases. However, much the same can be said for the 4 putative biomarkers mentioned above. FGF21 has been most studied and does appear to be related in some way to OXPHOS disorders. However, neither it nor the other listed compounds have been studied rigorously in randomized controlled trials (RCTs) of primary mt diseases sufficiently to allow the FDA or similar regulatory agencies to accept any as an a priori, valid biomarker that could be used in a RCT as a major outcome variable of a therapeutic intervention. The main objection is illustrated by the authors' own admission, summarized in lines 432-452, of the non-specificity of these molecules. As they state, many diseases, few (if any) of which are generally thought of as primarily acquired (much less congenital) mt diseases (eg., IBS, myocardial infarction), can elaborate 1 or more of these factors (FGF21, VEGF, etc.). Thus, what is their value as mt disease biomarkers when they can be shown to correlate with so many other conditions? 

          Again, we agree with the reviewer. However, we believe that even so, these parameters have a potential value as mitochondrial disease biomarkers and deserve to be studied. These are our arguments:

           In this review, we show how the secretion of different proteins is modified when there is a dysfunction of the oxidative phosphorylation system (OXPHOS). These observations suggest that the determination of the levels of these proteins in the blood of individuals suspected of suffering from OXPHOS diseases might be useful for the management of these patients. However, as mentioned in the manuscript ("Biosignatures” section), these changes are very unspecific, since their levels are also altered in other pathologies that are not directly related to the OXPHOS system. It is important to point out that some metabolites firmly established in the clinical investigation of mitochondrial diseases, like lactate or others, have a poor diagnostic sensitivity and specificity [1].

           Heterogeneity is a feature of virtually all parameters associated with OXPHOS diseases. These pathologies occur in men and women; at any age; with differences in clinical variables even for the same pathological mutation [2]; they can be multi-systemic or tissue-specific; histochemical (for example, ragged-red or cytochrome oxidase negative fibers) and biochemical markers (for example, drops in oxygen consumption or specific activities of the respiratory complexes) are not a common feature of all these pathologies either and can be observed in non-OXPHOS pathologies [3]. In any case, similar to the levels of secreted proteins discussed in our manuscript, none of the markers commented above can be considered a specific biomarker of OXPHOS disease. However, the levels of particular secreted proteins, given their ease of determination in blood, could be useful, together with other clinical clues, to determine if a patient suspected of suffering from one of these diseases should undergo a more invasive procedure, such as a muscle biopsy to study histochemical and biochemical markers of OXPHOS disease [4]. The study of these proteins could also help to determine which tissues may or may not be involved [5]. And then, depending on the affected tissues, it would help to select the most suitable one for the study of subsequent OXPHOS disease markers, like oxygen consumption or respiratory complex specific activities. Sometimes an OXPHOS disorder does not show clinical, histological or biochemical manifestations in the muscle, and the biopsy might have been avoidable [6]. In addition, the determination of the secreted protein levels could be useful, in patients already diagnosed with an OXPHOS disease, to predict their prognosis and response to treatment [1]. Since the secretion of several proteins is modified when there is an OXPHOS dysfunction, and there are no reports on the secretome of patients with OXPHOS diseases, studies on the secretome of patients with confirmed OXPHOS disease and control individuals should be carried out. These studies should also include patients with non-OXPHOS diseases with major involvement of the muscle and/or nervous system, which are the most affected tissues in mitochondrial diseases. Moreover, these analyses should consider different technical approaches to try to detect the largest number of proteins whose secretion is modified when there is an OXPHOS dysfunction. Then, after rigorous studies, the FDA or other regulatory agencies would be in conditions to propose or reject some of these proteins as valid biomarkers.

We have extended now the “Conclusions” section to include these arguments and emphasize the need for further studies in wider cohorts of patients with different mitochondrial diseases before diagnostic use. Lines 483-502.

[1] Monitoring clinical progression with mitochondrial disease biomarkers. Steele HE, Horvath R, Lyon JJ, Chinnery PF. Brain 2017; 140: 2530-2540.

[2] Resolving complexity in mitochondrial disease: Towards precision medicine. Boggan RM, Lim A, Taylor RW, McFarland R, Pickett SJ 2019. Molecular Genetics and Metabolism 2019; 128: 19-29.

[3] Biomarkers for mitochondrial energy metabolism diseases. Boenzi S, Diodato S. Essays in Biochemistry 2018; 62: 443-454.

[4] Mitochondrial disease: Advances in clinical diagnosis, management, therapeutic development, and preventive strategies. Muraresku CC, McCormick EM, Falk MJ. Current Genetic Medicine Reports 2018; 6: 62-72.

[5] Biomarkers for mitochondrial respiratory chain disorders. Suomalainen A. Journal of Inherited and Metabolic Disease 2011; 34: 277-282.

[6] Epilepsy due to mutations in the mitochondrial polymerase gamma (POLG) gene: A clinical and molecular genetic review. Anagnostou ME, Ng YS, Taylor RW, McFarland R. Epilepsia 2016; 57: 1531-1545.

Specific comments:

  1. The paper's Introduction is ~50% too long; it currently takes4 pages and over 140 lines to reach the beginning of the title's real theme: "OXPHOS dysfunction modifies the protein secretion by the cells".

We have largely reduced now the “Mitochondrial function and organization”, “Oxidative phosphorylation system”, “Mitochondrial DNA” and “Nuclear DNA” sections and fused them in only one section entitled “Oxidative phosphorylation system”. Lines 57-127.

  1. 1 includes various OXPHOS inhibitors, several of which are discuss later in the paper. It would be useful to refer back to this figure to remind the reader where these poisons act. 

We have now included references to this figure when OXPHOS inhibitors are mentioned on the paper. Lines 78, 87, 161, 187, 292, 344, 389, 420.

  1. Lines 123-124: this sentence reflects a tautology and can be eliminated. 

We have removed this sentence.

  1. line 125: a specific reference is required to justify this claim.

We have included a specific reference.

 10. Gorman, G.S.; Chinnery, P.F.; DiMauro, S.; Hirano, M.; Koga, Y.; McFarland, R.; Suomalainen, A.; Thorburn, D.R.; Zeviani, M.; Turnbull, D.M. Mitochondrial diseases. Nat. Rev. Dis. Prim. 2016, 2, 1-22.

  1. lines 226-238: what is the relevance of this passage to the article's main theme? It can be severely shortened or eliminated.

As commented in the manuscript, besides this study [1] discussed in this paragraph, and that from [2], no other secretomic analyses related to OXPHOS dysfunction have been published so far, and we wanted to emphasize it. In this study, authors gave an inducer of the mitochondrial reactive oxygen species (ROS) production, that also decreases the levels of subunits of respiratory complex I and II, to human stem-cell derived retinal pigmented epithelium. As mitochondrial ROS production is a frequent finding in OXPHOS disorders, this study could provide clues on the secretome of cells with OXPHOS dysfunction. In any case, we have widely shortened this paragraph. Lines 232-250.

[1] Proteome and secretome dynamics of human retinal pigment epithelium in response to reactive oxygen species. Meyer JG, García TY, Schilling B, Gibson BW, Lamba DA. Scientific Reports 2019; 9: 1-12.

[2] Pharmacologic concentrations of linezolid modify oxidative phosphorylation function and adipocyte secretome. Llobet L, Bayona-Bafaluy MP, Pacheu-Grau D, Torres-Pérez E, Arbones-Mainar JM, Navarro MA, Gómez-Díaz C, Montoya J, López-Gallardo E, Ruiz-Pesini E. Redox Biology 2017; 13: 244-254.

  1. line 278: some of these compounds are not identified in Fig. 1.

Carbony cyanide-p-trifluoromethoxyphenylhydrazone (FCCP) is the only compound that was not identified in Figure 1. This is not an inhibitor of any respiratory complex but an uncoupling agent. This compound has now been included in Figure 1 and Line 103-104.

  1. lines 316-17: either eliminate the word "remarkable" or defend its use with actual data.

In a recent study, sodium pyruvate significantly reduced plasma lactate, lactate/pyruvate ratio, serum GDF15 and lateral ventricular lactate in adult patients with genetically, biochemically and clinically confirmed mitochondrial disease. We have now added this reference to the manuscript. Line 329.

[1] Biomarkers and clinical rating scales for sodium pyruvate therapy in patients with mitochondrial disease. Koga Y, Povalko N, Inoue E, Nashiki K, Tanaka M. Mitochondrion 2019; 48: 11-15.

  1. lines 353-354: this statement should either be defended with data and references or eliminated. To what are the authors referring as "currently used clinical markers and how has their utility been compared to that of FGF21 and/or GDF15?

Currently used clinical markers referred to conventional biomarkers such as lactate, pyruvate, creatine kinase, the lactate-pyruvate ratio [1-3] and alanine and organic acids [4]. The data showed that GDF15 was a more sensitive biomarker for mitochondrial disorders than conventional ones. Thus, “AUC analyses indicated that the sensitivity and specificity were significantly larger for GDF-15 than those for FGF-21, lactate, pyruvate, the L/P ratio, or CK” [2]; and, “Finally, when considering GDF-15 levels from control and mitochondrial disease groups in a multivariate logistic regression model that included possible con-founders (age, lactate:pyruvate, insulin, glucose, cholesterol, triglycerides, and FGF-21), GDF-15 was the best predictor of mitochondrial disease (p < 0.002)” [3]. GDF15 and FGF21 were both found more specific for children with mitochondrial disease when compared with lactate and pyruvate. “In fact, in all patient groups we found that a proportion of patients (between 16% and 29% depending on the group) had normal lactate levels but both GDF-15 and FGF-21 elevated. This suggests that these factors are more sensitive than lactate” [4].

The reviewer is right, we included these references at the end of the next sentence. We have now located these references in the right place and defined the “currently used clinical diagnostic markers”. Line 367-368.

[1] Biomarkers for mitochondrial respiratory chain disorders. Suomalainen A. Journal of Inherited and Metabolic Disease 2011; 34: 277-282.

[2] Growth differentiation factor 15 as a useful biomarker for mitochondrial disorders. Yatsuga S, Fujita Y, Ishii A, Fukumoto Y, Arahata H, Kakuma T, Kojima T, Ito M, Tanaka M, Saiki R, Koga Y. Annals of Neurology 2015; 78: 814-823.

[3] A comparison of current serum biomarkers as diagnostic indicators of mitochondrial diseases. Davis RL, Liang C, Sue CM. Neurology 2016; 86: 2010-2015.

[4] GDF-15 is elevated in children with mitochondrial diseases and is induced by mitochondrial dysfunction. Montero R, Yubero D, Villarroya J, Henares D, Jou C, Rodriguez MA, Ramos F, Nascimento A, Ortez CI, Campistol J, Perez-Dueñas B, O’Callaghan M, Pineda M, Garcia-Cazorla A, Oferil JC, Montoya J, Ruiz-Pesini E, Emperador S, Meznaric M, Campderros L, Kalko SG, Villarroya F, Artuch R, Jimenez-Mallebrera C. PLoS One 2016; 11: 1-15.

  1. lines 357-391: it seems extremely unlikely, and misleading, to promote VEGF as a future biomarker of OXPHOS dysfunction. A major HIF1a -induced transcript, VEGF is increased in so many physiological and pathological conditions as to make it a sensitive and specific marker, alone or with other molecules, of mt diseases extremely unlikely. A similar concern applies to IL-6. These concerns beg the question how the authors would design a study to determine, with confidence, the validity of using the 4 biomarkers listed in the figure for the diagnosis of diseases of OXPHOS disorders.

We do not pretend to introduce VEGF or IL6 as sensitive and specific markers for OXPHOS disorders. However, as already commented in our second response to this reviewer, we believe they could have potential interest in different scenarios, such as to determine if a patient suspected of suffering from one of these diseases should undergo a more invasive procedure, such as a muscle biopsy to study histochemical and biochemical markers of OXPHOS disease; to determine which tissues may or may not be involved; and, in patients already diagnosed with an OXPHOS disease, to predict their prognosis and response to treatment.This argument has been now included in the chapter Biosignatures, lines 448-451, and Conclusions, lines 483-502.

        10. Fig 2: VEGF is misspelled.

We have corrected it now. Figure 2.

Submission Date 18 March 2020 Date of this review 23 Mar 2020 18:47:31

Reviewer 3 Report

Dear Editor

The review submitted by Garrido-Pérez and Co. is a collective and informative review of available knowledge about the role of mitochondrial dysfunction on the cellular secretome.

The review contains a broad range of information about cellular secretome and its clinical application.

I have only minor suggestions to make the current draft more attractive for readers of IJMS.

The role of mitochondria in ageing has been well studied, the link between defective mitochondria and cellular senescence needs to be discussed in the context of the review.

It would be great if authors include information about:

1. The mitochondrial dysfunction-associated senescence (Midas), since it is known that the composition of the secretome of Midas cells differs from normal cellular SASP.

2. The mitochondrial derived extracellular vesicles and exosome.

The novel information about the composition of secreted molecules as well as the mechanism of packing and releasing can be briefly mentioned.

The prognostic and therapeutic role of both released biomarkers can be discussed.

Sincerely

Author Response

Response to the reviewer 3

Dear Editor

The review submitted by Garrido-Pérez and Co. is a collective and informative review of available knowledge about the role of mitochondrial dysfunction on the cellular secretome.

The review contains a broad range of information about cellular secretome and its clinical application.

          We would like to thank this reviewer for his/her kind comments and interesting suggestions.

I have only minor suggestions to make the current draft more attractive for readers of IJMS.

The role of mitochondria in ageing has been well studied, the link between defective mitochondria and cellular senescence needs to be discussed in the context of the review.

It would be great if authors include information about:

  1. The mitochondrial dysfunction-associated senescence (Midas), since it is known that the composition of the secretome of Midas cells differs from normal cellular SASP.
  2. The mitochondrial derived extracellular vesicles and exosome.

The novel information about the composition of secreted molecules as well as the mechanism of packing and releasing can be briefly mentioned.

The prognostic and therapeutic role of both released biomarkers can be discussed.

            Ageing and cellular senescence are very interesting topics but we wanted to focus our review on OXHOS dysfunction and its effect on the secretome in patients suffering primary OXPHOS disorders.

            To our knowledge, a relationship between OXPHOS dysfunction and levels of secreted proteins in Midas is only reported in [1]. In fact, using the terms “Midas”, “mitochondria” and “secretome” in PubMed, we only recovered one publication [2], a comment on the article [1]. We previously considered the article [1] for our review. However, there is only one figure showing the effect of OXPHOS inhibitors on a secreted protein (IL6), and this is on irradiated cells (Figure S2D). Therefore, we believed that so few data did not merit the long description required to explain Midas and SASP phenotypes, when this was not our main interest for this review.

         Similarly, by considering the terms “exosome” and mitochondria”, we have not recovered any article on OXPHOS dysfunction in patients suffering from primary OXPHOS disorders but the reviewer has provided us with an excellent idea for a potential future revision on exosomes in primary OXPHOS disorders.

[1] Mitochondrial dysfunction induces senescence with a distinct secretory phenotype. Wiley CD, Velarde MC, Lecot P, Liu S, Sarnoski EA, Freund A, Shirakawa K, Lim HW, Davis SS, Ramanathan A, Gerencser AA, Verdin E, Campisi J. Cell Metabolism 2016; 23: 303-314.

[2] Mitochondrial dysfunction meets senescence. Gallage S, Gil J. Trends in Biochemical Sciences 2016; 41: 207-209.

Sincerely

Submission Date 18 March 2020 Date of this review 28 Mar 2020 18:29:23

Reviewer 4 Report

Oxidative phosphorylation dysfunction modifies the cell secretome

Garrido-Perez, et al.

Major comment:

This is an informative review on recent progress in biomarker studies for diagnosis of mitochondrial diseases.

Minor comments:

#1, Expression of GDF15 is increased by lactate and is decreased by pyruvate in cybrid cells carrying m.3243A>G.

Fujita et al. [46] reported the comparative transcriptomic study of the 2SD cybrid cells carrying m.3243A>G mutation exposed to high lactate or pyruvate and they subsequently surveyed proteins that are excreted from the cybrid cells and the serum from patients with mitochondrial diseases [46]. This paper was reported before the clinical study on GDF15 and FGF21 by Yatsuga et al. [47].

#2, Legend for Figure 1

For Cyt c and cytochrome c, “c” should be italicized.

“Reactive Oxygen Species” should be not be capitalized; “reactive oxygen species”

“Rotenone, Capsaicin, Antimycin A, Sodium cyanide, Hydrogen sulfide. Sodium azide, Oligomycin” should not be capitalized as well; “rotenone, capsaicin, antimycin A, sodium cyanide, hydrogen sulfide sodium azide, oligomycin”

#3, Figure 2

“Signaling sended to other organs” should read

“Signaling sent to other organs”

Author Response

Major comment:

This is an informative review on recent progress in biomarker studies for diagnosis of mitochondrial diseases.

We would like to thank this reviewer for his/her kind comments and interesting suggestions.

Minor comments:

#1, Expression of GDF15 is increased by lactate and is decreased by pyruvate in cybrid cells carrying m.3243A>G.

Fujita et al. [46] reported the comparative transcriptomic study of the 2SD cybrid cells carrying m.3243A>G mutation exposed to high lactate or pyruvate and they subsequently surveyed proteins that are excreted from the cybrid cells and the serum from patients with mitochondrial diseases [46]. This paper was reported before the clinical study on GDF15 and FGF21 by Yatsuga et al. [47].

We have now organized this section so the papers are cited more properly in the paragraph. Lines 318-335.

#2, Legend for Figure 1

For Cyt c and cytochrome c, “c” should be italicized.

We have corrected now this typo. Line 96 and Figure 1.

“Reactive Oxygen Species” should be not be capitalized; “reactive oxygen species”.

We have corrected it now. Line 101.

“Rotenone, Capsaicin, Antimycin A, Sodium cyanide, Hydrogen sulfide. Sodium azide, Oligomycin” should not be capitalized as well; “rotenone, capsaicin, antimycin A, sodium cyanide, hydrogen sulfide sodium azide, oligomycin”

 We have corrected it now. Line 101-104

#3, Figure 2

“Signaling sended to other organs” should read

“Signaling sent to other organs”

We have corrected it now. Line 484.

Round 2

Reviewer 2 Report

The authors and I retain a fundamental philosophical divide as to the overall value of their review, specifically in assigning any real import to the usefulness of the biomarkers cited in the diagnosis of primary mt diseases. Their defense is so qualified with words such as "may" or "might" as to seriously mitigate the probative value of their work, in my opinion. I do not accept their arguments that such molecules as VEGF may, at this juncture, have any discriminating value, alone or as some ill-defined collection of such compounds. The authors provide no logical, defensible strategy to synthesize from the wealth of data they summarize a logical, and testable proposal to move the field forward in rigorously evaluating the diagnostic worth of any of the putative biomarkers they cite.

Author Response

We would like to thank this reviewer for his/her effort in a deep analysis of our manuscript and we would like to emphasize that we agree with him/her in many of their appreciations. However, we don´t agree with this reviewer about our review overall value and we are sorry that we did not convince him/her about this point.

Our review proposes that dysfunction of the oxidative phosphorylation system (OXPHOS) can modify the secretion of certain proteins so the study of the secretome of patients with diseases due to mutations in genes related to the OXPHOS system could be useful in diagnosis, prognosis, treatment and monitoring of these patients.              

We would like to thank this reviewer as well for having valued our extensive bibliography analysis. Unfortunately, there is no studies on the secretome of OXPHOS patients. Therefore, to support our proposal, we rely on specific proteins whose secretion has already been found to be modify in these pathologies. The study of these proteins has already provided interesting information on patients with OXPHOS diseases, such as what kind of mutations modify their secretion to a greater extent. We believe this knowledge could be very useful in the prognosis, treatment and monitoring of the patients. On the other hand, another observation coming out from the study of these proteins is that of their non-specificity. Many other non-mitochondrial conditions also modify its secretion and this can be a handicap in their use for OXPHOS disease diagnosis.              

We agree the proteins that we discuss may not have any discriminatory value currently in the diagnosis of OXPHOS diseases, but in other aspects of patients already diagnosed with these diseases. However, in order to progress in this field, studies on secretomes need to be carried out and there would be no reason to write an article encouraging researchers to carry out these studies if their value has already been proved or discarded.              

Finally, when we submitted the article for the first time, we already agreed that the reviewers' comments would be published anonymously. Thus, we think that his/her point of view, along with that of the other three reviewers, could enrich the discussion on this field.